**Data Availability Statement:** The data file is enclosed in S1 Data.

**Funding:** The authors received no specific funding for this work.

# Predictors of health-related quality of Life for COVID-19 survivors living in Dhaka, Bangladesh: A repeated Follow-Up after 18 months of their recovery

**Md. Utba Rashid**[1,2], **Koustuv Dalal**[3]*, **Md. Abdullah Saeed Khan**[4,5],
**Umme Kulsum Monisha**[4,6], **Soumik Kha Sagar**[2,4], **Tahmina Zerin Mishu**[4,6],
**Mohammad Hayatun Nabi**[4], **Mohammad Delwer Hossain Hawlader**[4]

**1** Department of Epidemiology and Biostatistics, Arnold School of Public Health, University of South Carolina, Columbia, South Carolina, United States of America, **2** Nutrition Research Division (NRD), International Centre for Diarrhoeal Disease Research, Bangladesh, Mohakhali, Dhaka, Bangladesh, **3** School of Health Sciences, Division of Public Health Science, Mid Sweden University, Sundsvall, Sweden, **4** Department of Public Health, North South University, Bashundhara, Dhaka, Bangladesh, **5** National Institute of Preventive and Social Medicine (NIPSOM), Mohakhali, Dhaka, Bangladesh, **6** Public Health Promotion and Development Society (PPDS), Dhaka, Bangladesh

* koustuv.dalal@miun.se, koustuv2010@hotmail.com

## Abstract

The study aimed to explore the key determinants that impact the quality of life (QoL) transformation of those who have recuperated from COVID-19 in the Dhaka metropolis, particularly 18 months post-recovery. RT-PCR confirmed that 1,587 COVID-19 patients from Dhaka were included in the study. The baseline was June - November 2020, subsequently recovered and interviewed 18 months after their initial recovery. The follow-up included 1587 individuals using the WHOQOL-BREF questionnaire. After excluding 18 deaths, 53 refusals, 294 inaccessible participants, and 05 incomplete data entries, we analysed the data of the 1217 respondents. The average physical domain score decreased significantly from baseline to follow-up, whereas a significant increase in average scores has been observed in other domains at the follow-up (p<0.05). GEE equation shows after adjusting for other factors, older age, female gender, higher education, higher family income, hospital admission during COVID-19, and the number of comorbidities were significantly associated with changing individuals' QoL scores. Monthly family income >60000 BDT, being married and having no previous vaccination history are significant in reducing people's QoL scores in the psychological domain. On the other hand, age, employment status, monthly family income, marital status, smoking history, and COVID-19 reinfection were significantly associated with altering an individual's QoL scores in the social domain. The overall QoL of COVID-19 recovered people improved in all domains after 18 months, except the physical realm. Participants' age, employment status, family income, marital status, smoking history, comorbidities, COVID-19 vaccination, and COVID-19 reinfection were responsible for altering people's QoL index.

**Competing interests:** The authors have declared
that no competing interests exist.

## Introduction

The novel coronavirus (SARS-CoV-2) inflicted a pernicious blow on the global healthcare systems [1]. The rapidly spreading virus led governments to adopt social distancing measures, order prolonged lockdown periods, and implement strategies to prevent and restrict the quality of living [2]. The COVID-19 pandemic posed a grave threat to global public health, inducing an unprecedented level of anxiety, stress, apprehension, and discontent that has profoundly impacted Health-Related Quality of Life (HRQOL) on a global scale [3].

The city of Dhaka, the capital of Bangladesh, has been grappling with the COVID-19 crisis on all fronts since the onset of the pandemic in March 2020 [4]. To date, 1.25 million confirmed cases have been diagnosed in Dhaka alone, with the highest mortality rate (46%) among the various administrative divisions [5]. Furthermore, the healthcare system in the capital has been facing difficulty managing the high load of cases due to the scarcity of skilled personnel and inadequate resources [6]. The Government of Bangladesh took necessary steps to address the gaps [7]. However, the limited number of intensive care unit (ICU) beds made managing the severe cases difficult [5]. Furthermore, the lack of testing kits, erroneous testing results, insufficient medical equipment for healthcare personnel, and the absence of ICU beds with ventilation support for emergency patients have made it even more challenging for the government to contain the spread of the disease [4, 8]. These phenomena put the rehabilitation process of COVID-19-affected individuals in jeopardy and have a long-term impact on their well-being.

In recent times, Health-Related Quality of Life (HRQoL) (often interchangeably used with the term Quality of Life [QoL]) has been predominantly investigated in patients suffering from non-communicable and communicable diseases. The instrument made by the World Health Organization (WHO) to assess Quality of Life (QoL) HRQoL is a dynamic, subjective, and multi-faceted construct encompassing physical, social, psychological, and environmental domains [9, 10]. WHO conceptualises HRQoL as an individual's perception of physical fitness and fitness-associated domains [11]. COVID-19 infection has been found to cause a decline in HRQoL compared to the pre-COVID-19 times [12]. Additionally, severe COVID-19 increases the risk of developing physical illnesses like cardiovascular and respiratory diseases [13, 14] and long-term persistent specific symptoms, further lowering HRQoL [15].

Furthermore, studies conducted in Wuhan, China, the outbreak's epicentre, have demonstrated that almost half of the hospitalised patients experienced non-specific symptoms, such as respiratory difficulties, three months post-discharge [16]. Previous coronavirus outbreaks, such as severe acute respiratory syndrome (SARS) outbreak, had been shown to lower QoL of sufferers up to two years post-infection [17, 18].

COVID-19 was found to impair HRQoL in patients for a long time after recovery. Some even suggest differences in HRQoL by the severity of illness (19,20). The study in Amsterdam, Netherlands, described that people with mild coronavirus disease 2019 (COVID-19) experience better HRQoL after one year of disease onset than those with moderate or severe COVID-19 [19]. Despite numerous studies in the wake of the SARS-CoV-2 outbreak, a few tracked the changes in QoL over time among patients who recovered from COVID-19. As the Bangladeshi populace is currently going through a recovery phase from the aftermath of COVID-19 and Dhaka is the most prominent city where significant variation persists due to the migration of a vast population from the different corners of the country, evaluating the pattern of HRQoL over a substantial period can give us essential information for designing a proper rehabilitation package to improve the HRQoL of affected individuals during the post-

COVID-19 era. However, a few studies explored the long-term QoL of a person after recovery from COVID-19. This study aimed to identify the changes in QoL between two different post-COVID periods in the capital city of Dhaka, Bangladesh.

## Methodology

### Study design and study participants

The subjects of this post-recovery study were individuals diagnosed with COVID-19 using Reverse Transcription-Polymerase Chain Reaction (RT-PCR), living in Dhaka city between June 2020 and November 2020. Between November 2020 and January 2021, a baseline cross-sectional assessment of QoL was conducted among 1,587 convalescent COVID-19 participants from Dhaka city, utilising the WHOQOL-BREF instrument [20]. The detailed procedural methodology of the investigation was elaborated elsewhere [21]. Our focus was directed toward all respondents who partook in the baseline inquiry for this follow-up assessment. We excluded- (1) individuals who had died before the follow-up examination; (2) individuals who refused to participate; and (3) those who could not be contacted via telephone (owing to dis-connection, call waiting, an inactive line, or network problems). The follow-up investigation involved interviews with 1,222 participants after 18 subjects had deceased, 53 individuals had declined participation, and 294 people remained out of reach during the survey period (**S1 Fig**). Finally, after excluding 05 incomplete data, we analysed 1217 participants' HRQoL index in our current study. The follow-up survey was administered from mid-November 2021 to late January 2022, exactly one year after the baseline evaluation.

### Data collection procedure

We collected our survey data through the semi-structured questionnaire designed during the primary inquiry (1st interview) with minor alterations. Upon completing the revised question-naire, we disseminated the names of registered participants to our appointed data collectors. Our collectors contacted everyone on the roster to inquire about their availability for the follow-up interview. Upon obtaining verbal informed consent, the selected data collectors recorded information from the participants. Before acquiring the follow-up data, the data collectors assured the respondents that specific questions could be bypassed if they felt hesitant to answer.

### Study instrument

The antecedently checked structured questionnaire, employed during the primary interview, underwent minor revisions to incorporate a pair of inquiries related to the immunisation against COVID-19 and the frequency of reoccurrence between the initial and secondary interviews. The standard questionnaire comprised a sociodemographic summary, personal habits, comorbidities, COVID-19 immunisation, and reinfection.

### WHOQOL-BREF

We utilised WHOQOL-BREF, a succinct and validated rendition of the WHOQOL-100 quality of life assessment questionnaire [20], to evaluate the QoL of COVID-19 convalescent patients of Dhaka city in line with our previous study. The WHOQOL Consortium partnered with fifteen foreign field centres to conceive the latter instrument, engendering a QoL evaluation that could be implemented across divergent cultures. This 26-item QOL instrument has demonstrated commendable to outstanding psychometric properties and is interculturally responsive. The WHOQOL-BREF engenders a profile and score for each of the four QOL

domains; inquiries are oriented around respondents' significance to each facet of life and how challenging or satisfactory they perceive them. The Physical Health domain inquiries are based on quotidian activities, medical assistance, vitality, mobility, the degree of distress, sleep pattern, and work capacity. The Psychological domain is directed toward participants' convictions, emotions, self-esteem, cognitive faculties, and erudition. The Social Relationships domain probes into the respondent's overall contentment with their personal and social life. Finally, the environment domain comprises questions on security and safety, gratification with one's property and physical milieu, pecuniary resources (whether one has enough finances to meet one's needs), accessibility to requisite care, information, and transportation. Additionally, the questionnaire has two specific interrogatives regarding participants' perspectives on their general QoL and health. We utilised the Bangla-validated version of the original WHOQOL-BREF questionnaire [22].

## Ethics statement

The ethical review committee (ERC)/institutional review board (IRB) of North South University provided ethical approval for this project (2020/OR-NSU/IRB-No.0801). All procedures were carried out per the Helsinki Declaration of 1964 and subsequent revisions or comparable ethical norms. Informed verbal consent was taken before the inclusion of the study participants at the beginning of each telephone interview, approved by the ethical committee.

## Statistical analysis

We transformed the WHOQOL-BREF scores into a 100-point scale as stipulated by the calculation guidelines [20, 22]. The descriptive statistics were displayed in terms of frequency (percentage) or mean (±standard deviation). A paired-sample t-test was conducted to appraise the modifications in QoL scores over one year, comparing the scores between the first and second interviews. Moreover, we employed an independent sample t-test or Analysis of Variance (ANOVA) for each interview point to compare QoL scores across the groups of factor variables. To ascertain differences in QoL scores across categories of independent variables after adjusting for intra-individual variation between two interview points, we employed a Generalized Estimating Equation (GEE) analysis. We assessed whether the score increased, decreased, or remained unchanged for each patient in the four domains. Additionally, using multivariable logistic regression, we separately analysed the factors leading to a decline in QoL scores from the first to the second interview in the four domains. The statistical tests were conducted on the Stata version 16 software, and the graphs were created using the statistical software R Studio (version 2022.07.1) and Microsoft Excel Version 2019.

## Results

The sociodemographic factors and comorbidities of the study participants are shown in **Table 1** (n = 1217). The majority of the participants were aged 46 years or above (27.53%), male (65.41%), urban dwellers (97.78%), graduated (42.65%), service holders (56.86%), married (81.10%), and having a monthly family income of 20001–40000 BDT (37.14%). A considerable improvement was evidenced in participants' QoL among all the domains except for the physical domain. We also observed no significant improvement in individuals' overall perception regarding their QoL between the two follow-ups. Instead, a slight deterioration was detected in their overall satisfaction with their health (as measured by Q1 and Q2) (**Fig 1**). Between the two interviews, the physical domain scores significantly declined, while the psychological and environmental domain scores improved remarkably (p<0.05) (**S2 Fig**).

**Table 1. Socio–demographic profile and comorbidities of the participants (N = 1217).**

| Characteristics | Category | Frequency (n) | Percentage (%) |
|---|---|---|---|
| **Age** | <26 | 115 | 9.45 |
| | 26–30 | 236 | 19.39 |
| | 31–35 | 204 | 16.76 |
| | 36–40 | 199 | 16.35 |
| | 41–45 | 128 | 10.52 |
| | 46+ | 335 | 27.53 |
| **Gender** | Male | 796 | 65.41 |
| | Female | 421 | 34.59 |
| **Residence** | Rural | 6 | 0.49 |
| | Urban | 1190 | 97.78 |
| | Semi-urban | 21 | 1.73 |
| **Educational Status** | No formal education | 11 | 0.90 |
| | Primary | 50 | 4.11 |
| | Up to SSC | 113 | 9.29 |
| | Up to HSC | 250 | 20.54 |
| | Graduation | 519 | 42.65 |
| | Post-graduation | 274 | 22.51 |
| **Employment Status** | Service | 692 | 56.86 |
| | Business | 194 | 15.94 |
| | Farmer | 2 | 0.16 |
| | Housewife | 178 | 14.63 |
| | Student | 66 | 5.42 |
| | Unemployed | 46 | 3.78 |
| | Others | 39 | 3.20 |
| **Monthly Family Income in BDT** | ≤ 20000 | 197 | 16.19 |
| | 20001–40000 | 452 | 37.14 |
| | 40001–60000 | 286 | 23.50 |
| | >60000 | 282 | 23.17 |
| **Marital Status** | Single | 190 | 15.61 |
| | Married | 987 | 81.10 |
| | Separated | 3 | 0.25 |
| | Divorced | 12 | 0.99 |
| | Widowed/Widower | 25 | 2.05 |
| **Health Care Worker** | No | 1049 | 86.20 |
| | Yes | 168 | 13.80 |
| **Smoking Status** | No | 826 | 67.87 |
| | Yes | 295 | 24.24 |
| | Past Smoker | 96 | 7.89 |
| **Hypertension** | No | 963 | 79.13 |
| | Yes | 254 | 20.87 |
| **Diabetes Mellitus** | No | 971 | 79.79 |
| | Yes | 246 | 20.21 |
| **Heart Diseases** | No | 1097 | 90.14 |
| | Yes | 120 | 9.86 |
| **Asthma** | No | 1022 | 83.98 |
| | Yes | 195 | 16.02 |

(*Continued*)

**Table 1.** (Continued)

| Characteristics | Category | Frequency (n) | Percentage (%) |
|---|---|---|---|
| **Chronic Kidney Disease (CKD)** | No | 1151 | 94.58 |
| | Yes | 66 | 5.42 |
| **Cancer** | No | 1159 | 95.23 |
| | Yes | 58 | 4.77 |

**Table 2** summarises the changes in the participants' QoL based on numerous factors throughout the multiple interviews. There was a noticeable reduction in the physical domain scores among respondents aged ≥26 years, urban/semi–urban tenants, educated, employed, and had a family income of >20000 BDT/month, regardless of sex, marital status, health care workers (HCWs), hospital admission, or smoking status. Contrastingly, the psychological domain score increased significantly in participants who were males, urban/semi–urban residents, employed, having a monthly family income of ≤40000 BDT, married, smokers, and with a history of hospital admission during the COVID–19 outbreak. Similarly, participants aged <36 years, males, rural residents, employed, single, smokers, and had <40000 BDT/ month family income showed a substantial rise in the social domain scores. However, a drastic

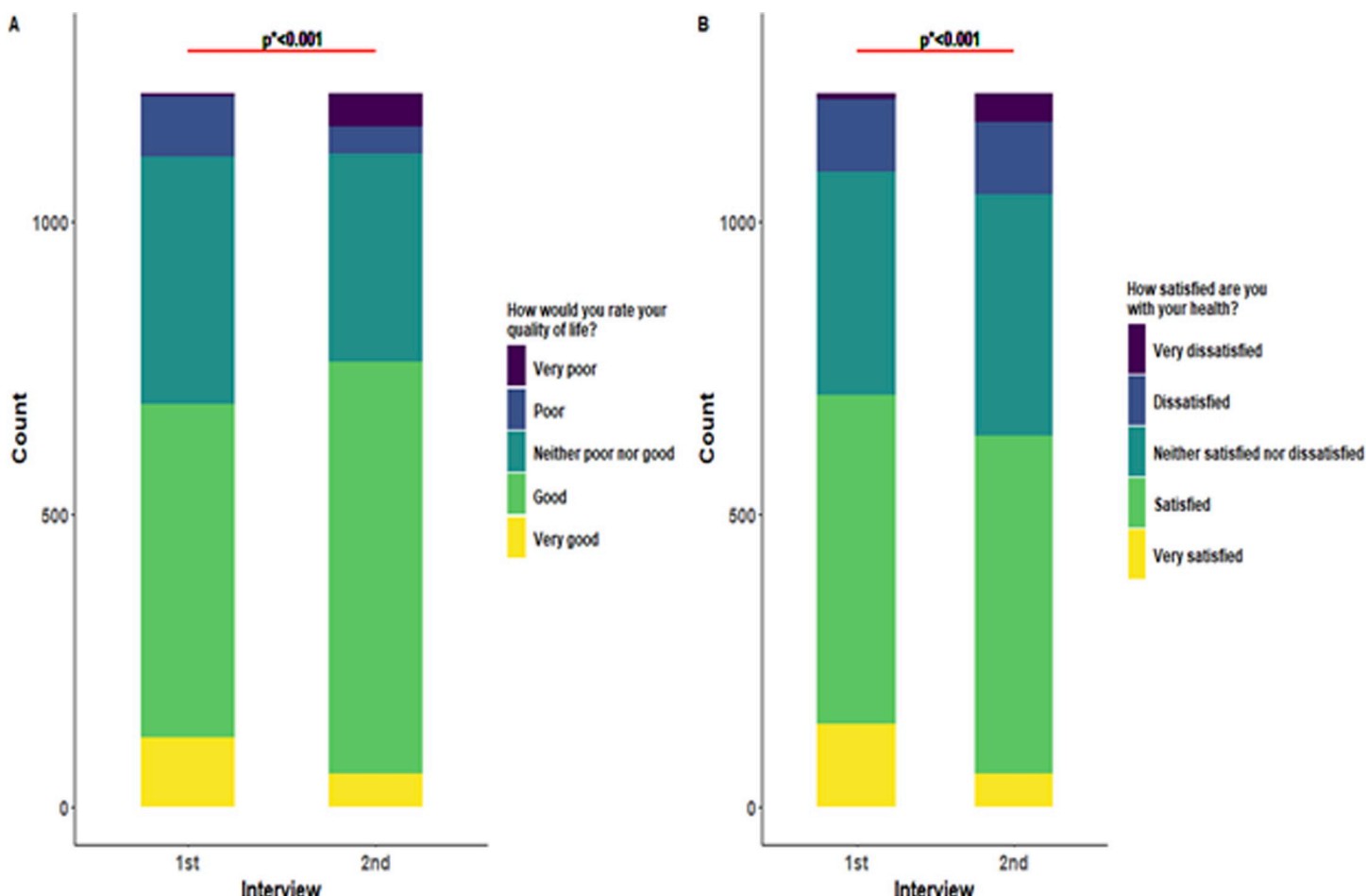

**Fig 1. Pattern of changes in overall quality of life and health satisfaction over the study period.**

**Table 2. Comparison of quality of life between baseline and follow-up interviews.**

| Variables | | Physical | | | Psychological | | | Social | | | Environmental | | |
|---|---|---|---|---|---|---|---|---|---|---|---|---|---|
| | | 1st visit | 2nd visit | p value | 1st visit | 2nd visit | p value | 1st visit | 2nd visit | p value | 1st visit | 2nd visit | p value |
| | | Mean (SD) | Mean (SD) | | Mean (SD) | Mean (SD) | | Mean (SD) | Mean (SD) | | Mean (SD) | Mean (SD) | |
| **Overall Domain Score** | | 67.87 (14.20) | 63.17 (13.37) | **<0.001** | 62.47 (15.55) | 63.76 (14.01) | **0.01** | 63.72 (19.42) | 64.15 (17.41) | 0.48 | 63.37 (12.54) | 66.44 (12.14) | **<0.001** |
| **Age** | <26 | 71.40 (14.42) | 68.70 (12.04) | 0.06 | 67.89 (14.45) | 68.78 (12.84) | 0.60 | 47.57 (17.87) | 52.96 (18.61) | **0.002** | 63.80 (12.85) | 68.03 (11.10) | **0.006** |
| | 26–35 | 70.48 (14.08) | 67.03 (12.70) | **<0.001** | 65.30 (15.24) | 66.71 (13.83) | 0.11 | 63.20 (21.01) [a] | 66.28 (17.99) [a] | **0.005** | 63.67 (13.16) | 67.38 (12.15) | **<0.001** |
| | 36–45 | 68.56 (13.55) | 61.01 (13.36) [ab] | **<0.001** | 60.37 (15.58) [ab] | 62.14 (14.46) [ab] | 0.08 | 70.05 (16.32) [ab] | 67.18 (15.14) [a] | **0.01** | 64.58 (11.65) | 65.61 (13.03) | 0.30 |
| | ≥46 | 62.57 (13.47) [abc] | 58.33 (12.51) [abc] | **<0.001** | 58.95 (15.15) [ab] | 59.74 (12.78) [ab] | 0.44 | 63.77 (17.16) [ac] | 62.26 (16.56) [abc] | 0.16 | 61.65 (12.31) [c] | 65.48 (11.15) | **<0.001** |
| | p-value | **<0.001** | **<0.001** | | **<0.001** | **<0.001** | | **<0.001** | **<0.001** | | **0.02** | **0.04** | |
| **Gender** | Male | 68.52 (14.18) | 64.15 (13.25) | **<0.001** | 63.63 (15.62) | 65.02 (14.08) | **0.04** | 62.87 (19.44) | 64.56 (17.08) | **0.03** | 63.42 (12.54) | 66.64 (11.88) | **<0.001** |
| | Female | 66.65 (14.18) | 61.33 (13.40) | **<0.001** | 60.29 (15.19) | 61.38 (13.57) | 0.20 | 65.32 (19.29) | 63.38 (18.01) | **0.05** | 63.29 (12.54) | 66.08 (12.63) | **0.002** |
| | p-value | **0.03** | **<0.001** | | **<0.001** | **<0.001** | | **0.04** | 0.26 | | 0.86 | 0.45 | |
| **Residence** | Rural | 67.83 (15.61) | 59.50 (11.71) | 0.25 | 59.33 (11.02) | 60.50 (10.97) | 0.83 | 51.00 (16.44) | 74.00 (2.45) | **0.02** | 65.67 (18.05) | 66.83 (8.45) | 0.91 |
| | Urban/ Semi-urban | 67.87 (14.20) | 63.19 (13.37) | **<0.001** | 62.49 (15.57) | 63.78 (14.02) | **0.01** | 63.78 (19.42) | 64.10 (17.44) | 0.60 | 63.36 (12.51) | 66.44 (12.16) | **<0.001** |
| | p-value | 0.99 | 0.50 | | 0.62 | 0.57 | | 0.11 | 0.16 | | 0.65 | 0.94 | |
| **Educational Status** | No or primary education | 63.61 (16.16) | 61.64 (15.05) | 0.40 | 55.28 (14.89) | 60.07 (13.44) | 0.05 | 56.72 (18.73) | 56.87 (18.65) | 0.96 | 60.30 (14.20) | 62.98 (11.33) | 0.26 |
| | Up to HSC | 66.46 (14.75) | 63.11 (13.51) | **<0.001** | 62.02 (15.40) [a] | 63.54 (14.48) | 0.13 | 61.27 (19.68) | 62.27 (17.71) | 0.35 | 62.47 (12.22) | 65.80 (12.25) | **<0.001** |
| | Graduation/ Above | 68.85 (13.68) [ab] | 63.32 (13.17) | **<0.001** | 63.24 (15.54) [a] | 64.15 (13.80) | 0.15 | 65.38 (19.13) [ab] | 65.58 (16.96) [ab] | 0.80 | 64.02 (12.50) [a] | 67.00 (12.11) [a] | **<0.001** |
| | p-value | **0.002** | 0.64 | | **<0.001** | 0.08 | | **<0.001** | **<0.001** | | **0.02** | **0.02** | |
| **Employment Status** | Unemployed | 67.46 (15.19) | 62.70 (13.92) | 0.07 | 62.67 (13.41) | 62.22 (13.17) | 0.85 | 57.89 (19.54) | 53.76 (19.30) | 0.20 | 62.98 (9.84) | 65.76 (10.87) | 0.22 |
| | Employed | 67.89 (14.17) | 63.19 (13.35) | **<0.001** | 62.47 (15.63) | 63.82 (14.04) | **0.01** | 63.95 (19.39) | 64.56 (17.21) | 0.32 | 63.39 (12.64) | 66.47 (12.19) | **<0.001** |
| | p-value | 0.84 | 0.81 | | 0.93 | 0.45 | | **0.04** | **<0.001** | | 0.83 | 0.70 | |
| **Monthly Family Income in BDT** | ≤20000 | 64.68 (15.55) | 62.41 (14.21) | 0.07 | 60.19 (16.92) | 64.36 (14.41) | **0.01** | 59.65 (21.38) | 63.38 (18.07) | **0.03** | 62.35 (15.32) | 65.60 (11.81) | **0.02** |
| | 20001–40000 | 67.15 (13.78) | 64.03 (12.89) | **<0.001** | 62.64 (14.63) | 64.87 (13.62) | **0.01** | 62.38 (18.47) | 64.88 (17.47) | **0.01** | 61.53 (12.05) | 66.06 (11.69) | **<0.001** |
| | 40001–60000 | 68.61 [a] (13.67) | 64.59 (14.09) | **<0.001** | 63.76 (15.20) | 64.43 (13.67) | 0.51 | 62.31 (19.61) | 64.10 (17.56) | 0.15 | 63.99 (12.28) | 67.29 (11.64) | **<0.001** |
| | >60000 | 70.50 [ab] (13.94) | 60.89 [bc] (12.46) | **<0.001** | 62.49 (16.25) | 60.90 [abc] (14.36) | 0.12 | 70.14 [abc] (17.83) | 63.58 (16.70) | **<0.001** | 66.41 [ab] (10.72) | 66.80 (13.51) | 0.70 |
| | p-value | **<0.001** | **0.003** | | 0.10 | **0.001** | | **<0.001** | 0.69 | | **<0.001** | 0.39 | |
| **Marital Status** | Single | 70.54 (15.40) | 67.59 (14.29) | **0.01** | 65.23 (15.27) | 66.66 (14.48) | 0.24 | 41.23 (13.71) | 50.67 (20.22) | **<0.001** | 63.07 (12.51) | 68.17 (12.10) | **<0.001** |
| | Married | 67.25 (13.84) | 62.14 (12.93) | **<0.001** | 61.83 (15.55) | 63.09 (13.82) | **0.03** | 68.96 (16.61) | 67.29 (15.05) | **0.01** | 63.44 (12.55) | 66.04 (12.12) | **<0.001** |
| | p-value | **0.002** | **<0.001** | | **0.003** | **<0.001** | | **<0.001** | **<0.001** | | 0.68 | **0.02** | |

(*Continued*)

**Table 2.** (Continued)

| Variables | | Physical | | | Psychological | | | Social | | | Environmental | | |
|---|---|---|---|---|---|---|---|---|---|---|---|---|---|
| | | 1st visit | 2nd visit | p value | 1st visit | 2nd visit | p value | 1st visit | 2nd visit | p value | 1st visit | 2nd visit | p value |
| | | Mean (SD) | Mean (SD) | | Mean (SD) | Mean (SD) | | Mean (SD) | Mean (SD) | | Mean (SD) | Mean (SD) | |
| **Health Care Worker** | No | 67.59 (14.37) | 63.05 (13.23) | **<0.001** | 62.59 (15.69) | 63.40 (14.00) | 0.16 | 63.37 (19.24) | 64.15 (17.40) | 0.24 | 63.28 (12.67) | 66.33 (12.11) | **<0.001** |
| | Yes | 69.64 (12.98) | 63.92 (14.19) | **<0.001** | 61.74 (14.69) | 66.00 (13.88) | **0.001** | 65.90 (20.44) | 64.19 (17.48) | 0.29 | 63.93 (11.71) | 67.17 (12.36) | **0.02** |
| | p-value | 0.08 | 0.44 | | 0.51 | **0.03** | | 0.12 | 0.98 | | 0.53 | 0.41 | |
| **Hospital Admission** | No | 70.19 (14.08) | 65.22 (12.60) | **<0.001** | 64.98 (14.98) | 57.01 (15.37) | 0.51 | 65.54 (19.74) | 65.78 (17.12) | 0.74 | 63.95 (12.42) | 67.12 (11.94) | **<0.001** |
| | Yes | 62.80 (13.11) | 58.71 (13.90) | **<0.001** | 65.39 (13.66) | 60.21 (14.10) | **<0.001** | 59.76 (18.11) | 60.61 (17.51) | 0.45 | 62.10 (12.72) | 64.97 (12.44) | **0.002** |
| | p-value | **<0.001** | **<0.001** | | **<0.001** | **<0.001** | | **<0.001** | **<0.001** | | **0.02** | **0.004** | |
| **Smoking Status** | No | 68.46 (14.21) | 63.14 (13.13) | **<0.001** | 63.36 (15.34) | 63.83 (14.01) | 0.45 | 64.74 (19.38) | 64.09 (17.55) | 0.36 | 63.91 (12.41) | 66.56 (11.85) | **<0.001** |
| | Yes | 66.08 (14.19) [a] | 62.85 (13.81) | **0.004** | 59.46 (15.82) [a] | 64.34 (14.14) | **<0.001** | 60.08 (19.38) [a] | 64.49 (16.94) | **<0.001** | 61.82 (13.19) [a] | 66.15 (12.78) | **<0.001** |
| | Past Smoker | 68.28 (13.86) | 64.41 (14.00) | **0.02** | 64.14 (15.47) [b] | 61.35 (13.46) | 0.15 | 66.14 (18.60) [b] | 63.64 (17.77) | 0.30 | 63.56 (11.26) | 66.40 (12.73) | 0.10 |
| | p-value | **0.04** | 0.61 | | **<0.001** | 0.19 | | **<0.001** | 0.90 | | **0.05** | 0.88 | |

Scores were expressed as Mean ±SD

[a-c] Scores with different superscript letters have a statistically significant difference across categories of the variable within a domain, e.g., values with a superscript 'a' is significantly different from value in the first category, 'b' is significantly different from the value in second category and 'c' is significantly different from the value in third category.

P-value was determined using one-way ANOVA with post-hoc analysis by Bonferroni multiple comparison test

decline was also shown in this domain among the interviewees aged 36–45 years, married, females, and having a monthly family income of >60000 BDT. Lastly, for the environmental domain, the average scores elevated significantly for all characteristics except participants aged 36–45 years, with no or primary education, unemployed, villagers, and had a smoking history with >60000 BDT/month family income.

Of all, 15.69% of participants developed one or more chronic diseases between the first and second interviews and 11.75% of participants were re-infected from COVID-19 during the follow-up period (S3 Fig).

During the follow-up, the physical domain's average QoL score dropped dramatically in the absence of any chronic diseases. On the contrary, there was an increase in the psychological domain's mean score in respondents with chronic diseases between two follow-up visits, except for chronic kidney disease (CKD) and cancer. Likewise, the social domain's average score was elevated only among the interviewees with heart disease or asthma. Except for heart disease, asthma, and cancer, the average QoL scores of the environmental domain improved significantly for the participants both with/without chronic diseases between baseline and follow-up (S1 Table).

Table 3 represents the correlation between QoL scores and associated factors using the generalised estimating equation model (GEE) after adjusting for other factors. The participants' age ≥36 years, female, hospital admission during the COVID-19 outbreak, and chronic diseases significantly related to the reduced physical and psychological QoL scores. Conversely, income (≥40001 BDT and ≥20001 BDT, respectively) was considerably connected to the

**Table 3. Factors associated with quality-of-life scores after adjusting for intra-individual changes between two interviews and for other factors by generalized estimating equation model (GEE).**

| Variables | | Difference in Physical Score | | Difference in Psychological Score | | Difference in Social Score | | Difference in Environmental Score | |
|---|---|---|---|---|---|---|---|---|---|
| | | Coefficient | p value | Coefficient | p value | Coefficient | p value | Coefficient | p value |
| **Age** | <26 (Ref) | 1 | | 1 | | 1 | | 1 | |
| | 26–35 | - 0.09 | 0.94 | - 1.94 | 0.11 | - 0.46 | 0.72 | - 0.61 | 0.61 |
| | 36–45 | - 2.49 | **0.05** | - 5.63 | **<0.001** | - 0.76 | 0.60 | - 0.85 | 0.49 |
| | ≥46 | - 4.63 | **0.001** | - 4.42 | **0.001** | - 0.89 | 0.53 | - 1.34 | 0.29 |
| **Gender** | Male (Ref) | 1 | | 1 | | 1 | | 1 | |
| | Female | - 2.35 | **<0.001** | - 3.69 | **<0.001** | 0.49 | 0.53 | - 0.15 | 0.80 |
| **Residence** | Rural (Ref) | 1 | | 1 | | 1 | | 1 | |
| | Urban/ Semi-urban | 2.61 | 0.24 | - 0.37 | 0.88 | 1.84 | 0.44 | - 0.87 | 0.76 |
| **Educational Status** | No or primary education (Ref) | 1 | | 1 | | 1 | | 1 | |
| | Up to HSC | 0.39 | 0.79 | 3.53 | **0.007** | 3.26 | 0.05 | 2.14 | 0.09 |
| | Graduation/ Above | 0.12 | 0.93 | 3.53 | **0.007** | 3.52 | **0.03** | 3.29 | **0.007** |
| **Employment Status** | Unemployed (Ref) | 1 | | 1 | | 1 | | 1 | |
| | Employed | - 0.19 | 0.91 | 2.30 | 0.15 | 0.43 | 0.77 | 1.33 | 0.35 |
| **Monthly Family Income in BDT** | ≤20000 (Ref) | 1 | | 1 | | 1 | | 1 | |
| | 20001–40000 | 1.23 | 0.14 | 2.08 | **0.02** | 1.16 | 0.25 | - 0.13 | 0.89 |
| | 40001–60000 | 2.19 | **0.01** | 2.19 | **0.02** | 1.76 | 0.10 | 1.67 | 0.07 |
| | >60000 | 3.70 | **<0.001** | 2.29 | **0.03** | 6.12 | **<0.001** | 3.08 | **0.001** |
| **Marital Status** | Single (Ref) | 1 | | 1 | | 1 | | 1 | |
| | Married | - 0.91 | 0.32 | 0.22 | 0.81 | 28.58 | **<0.001** | - 0.29 | 0.71 |
| **Health Care Worker** | No (Ref) | 1 | | 1 | | 1 | | 1 | |
| | Yes | 1.44 | 0.07 | 1.40 | 0.14 | 0.54 | 0.56 | 0.38 | 0.61 |
| **Hospital Admission** | No (Ref) | 1 | | 1 | | 1 | | 1 | |
| | Yes | - 3.98 | **<0.001** | - 4.20 | **<0.001** | - 4.72 | **<0.001** | - 1.26 | **0.03** |
| **Smoking Status** | No (Ref) | 1 | | 1 | | 1 | | 1 | |
| | Yes | - 0.40 | 0.55 | - 1.01 | 0.20 | - 1.42 | 0.07 | - 0.34 | 0.60 |
| | Past Smoker | 0.07 | 0.95 | - 0.97 | 0.40 | - 0.15 | 0.89 | - 0.74 | 0.44 |
| **Number of Chronic Disease** | 0 (Ref) | 1 | | 1 | | 1 | | 1 | |
| | 1 | - 3.99 | **<0.001** | - 2.25 | **0.003** | - 1.59 | **0.04** | - 0.48 | 0.46 |
| | 2 | - 6.81 | **<0.001** | - 5.05 | **<0.001** | - 4.28 | **<0.001** | - 0.37 | 0.70 |
| | ≥3 | - 8.06 | **<0.001** | - 6.93 | **<0.001** | - 5.67 | **<0.001** | - 2.26 | **0.02** |

elevated physical and psychological domain scores. Simultaneously, education was markedly associated with an increased psychological domain score only. Meanwhile, the lower social and environmental QoL scores were noticeably connected with hospital admission and chronic diseases, except for three or more comorbidities, which were substantially linked with a declined environmental QoL score only. Furthermore, higher education and higher income strongly correlated with higher social and environmental domain scores.

According to the multivariate logistic regression analysis (**Table 4**), the increasing age was dramatically linked to a decline in social domain scores. The employed respondents were 62% (AOR: 0.38, 95%CI: 0.19–0.79) less likely to encounter a reduced QoL in the social domain than the unemployed. Participants with an income of ≥60000 BDT/month experienced 1.70 (1.70, 1.13–2.54), 1.91 (1.91, 1.26–2.89), and 1.81 (1.81, 1.19–2.75) times deteriorated QoL than those with an income of ≤20,000 BDT in physical, psychological, and social domains, respectively. Besides, the married individuals were 1.70 (1.70, 1.15–2.52), 1.87 (1.87, 1.26–

**Table 4. Logistic regression model to identify factors that are associated with a decline in QoL score from baseline to follow-up interview.**

| Variables | | Physical | | Psychological | | Social | | Environmental | |
|---|---|---|---|---|---|---|---|---|---|
| | | Adj. OR | 95% CI | Adj. OR | 95% CI | Adj. OR | 95% CI | Adj. OR | 95% CI |
| **Age** | <26 (Ref) | 1 | | 1 | | 1 | | 1 | |
| | 26–35 | 1.03 | 0.62–1.70 | 0.89 | 0.52–1.51 | 1.85 | **1.02–3.36** | 0.76 | 0.44–1.31 |
| | 36–45 | 1.44 | 0.83–2.52 | 0.91 | 0.51–1.63 | 2.69 | **1.42–5.10** | 0.91 | 0.50–1.63 |
| | ≥46 | 1.50 | 0.86–2.64 | 1.20 | 0.67–2.16 | 2.98 | **1.56–5.69** | 0.60 | 0.33–1.10 |
| **Gender** | Male (Ref) | 1 | | 1 | | 1 | | 1 | |
| | Female | 0.93 | 0.70–1.25 | 1.09 | 0.82–1.47 | 1.21 | 0.90–1.63 | 1.12 | 0.83–1.52 |
| **Residence** | Rural (Ref) | 1 | | 1 | | 1 | | 1 | |
| | Urban/ Semi-urban | 0.60 | 0.09–4.15 | 1.12 | 0.20–6.28 | 3.00 | 0.33–27.66 | 1.02 | 0.18–5.74 |
| **Educational Status** | No or primary education (Ref) | 1 | | 1 | | 1 | | 1 | |
| | Up to HSC | 1.23 | 0.68–2.23 | 1.58 | 0.84–2.97 | 0.96 | 0.53–1.76 | 1.70 | 0.87–3.33 |
| | Graduation/ Above | 1.24 | 0.69–2.23 | 1.68 | 0.90–3.14 | 1.08 | 0.60–1.96 | 1.47 | 0.75–2.85 |
| **Employment Status** | Unemployed (Ref) | 1 | | 1 | | 1 | | 1 | |
| | Employed | 0.85 | 0.42–1.73 | 1.17 | 0.57–2.41 | 0.38 | **0.19–0.79** | 1.03 | 0.49–2.17 |
| **Monthly Family Income in BDT** | ≤20000 (Ref) | 1 | | 1 | | 1 | | 1 | |
| | 20001–40000 | 0.95 | 0.66–1.36 | 1.14 | 0.78–1.66 | 0.91 | 0.63–1.34 | 0.81 | 0.55–1.18 |
| | 40001–60000 | 1.20 | 0.80–1.78 | 1.26 | 0.83–1.92 | 1.08 | 0.72–1.65 | 0.99 | 0.65–1.50 |
| | >60000 | 1.70 | **1.13–2.54** | 1.91 | **1.26–2.89** | 1.81 | **1.19–2.75** | 1.42 | 0.94–2.15 |
| **Marital Status** | Single (Ref) | 1 | | 1 | | 1 | | 1 | |
| | Married | 1.39 | 0.97–2.01 | 1.70 | **1.15–2.52** | 1.87 | **1.26–2.79** | 1.83 | **1.21–2.78** |
| **Health Care Worker** | No (Ref) | 1 | | 1 | | 1 | | 1 | |
| | Yes | 1.12 | 0.77–1.61 | 0.70 | 0.48–1.03 | 1.20 | 0.83–1.75 | 1.21 | 0.90–1.61 |
| **Hospital Admission** | No (Ref) | 1 | | 1 | | 1 | | 1 | |
| | Yes | 1.22 | 0.92–1.63 | 0.87 | 0.65–1.16 | 1.08 | 0.81–1.45 | 1.21 | 0.90–1.61 |
| **Smoking Status** | No (Ref) | 1 | | 1 | | 1 | | 1 | |
| | Yes | 0.81 | 0.58–1.11 | 0.79 | 0.57–1.11 | 0.70 | **0.50–0.98** | 0.89 | 0.63–1.24 |
| | Past Smoker | 0.59 | **0.37–0.94** | 1.36 | 0.85–2.19 | 1.18 | 0.73–1.91 | 1.14 | 0.70–1.86 |
| **Number of Chronic Disease** | 0 (Ref) | 1 | | 1 | | 1 | | 1 | |
| | 1 | 0.53 | **0.39–0.73** | 0.84 | 0.61–1.15 | 0.74 | 0.54–1.01 | 1.02 | 0.74–1.40 |
| | 2 | 0.73 | 0.46–1.17 | 0.74 | 0.46–1.20 | 1.07 | 0.67–1.71 | 0.98 | 0.60–1.60 |
| | ≥3 | 0.38 | **0.24–0.59** | 0.68 | 0.42–1.09 | 0.13 | 0.01–1.46 | 0.87 | 0.54–1.40 |
| **New Chronic Disease** | No (Ref) | 1 | | 1 | | 1 | | 1 | |
| | Yes | 1.05 | 0.74–1.50 | 1.06 | 0.74–1.51 | 0.87 | 0.61–1.24 | 0.91 | 0.62–1.31 |
| **COVID-19 Vaccination** | Yes (Ref) | 1 | | 1 | | 1 | | 1 | |
| | No | 1.11 | 0.84–1.48 | 1.52 | **1.14–2.03** | 1.29 | 0.96–1.74 | 1.08 | 0.80–1.45 |
| **COVID-19 Re-infection** | No (Ref) | 1 | | 1 | | 1 | | 1 | |
| | Yes | 0.67 | 0.45–1.00 | 1.39 | 0.94–2.06 | 1.71 | **1.15–2.55** | 0.77 | 0.50–1.18 |

2.79), and 1.83 (1.83, 1.21–2.78)-fold raised likelihood to undergo the worsened psychological, social, and environmental QoL scores than the singles. Respondents with one and three or more chronic diseases were 47% (0.53, 0.39–0.73) and 62% (0.38, 0.24–0.59) less likely to have a diminished QoL in the physical domain, respectively, than those without chronic diseases. The absence of COVID–19 vaccination was related to 1.52 times (1.52, 1.14–2.03) more significant possibility of having a decreased psychological domain score than their counterparts between the interviews. Consequently, interviewees with a history of COVID-19 reinfection had a 1.71-fold (1.71, 1.15–2.55) increased likelihood of a lower social QoL score.

## Discussion

The pernicious impact of COVID-19 debilitated our way of existence and presented a pressing menace to our national economy from its inception. As a resource-poor country, the belated acceptance of appropriate intervention packages, in conjunction with limited disease knowledge, infodemic, insufficient laboratory support, limited personal protective equipment [23] and medical resources, the populace's attitude towards receiving vaccines [24, 25], and documented vaccine side effects [26, 27], were the principal factors that impeded our ability to control the virus during its initial stage. However, diligent research, nationwide awareness initiatives, and ultimately, mass vaccination aided in mitigating the situation, lowering the death toll, and eventually triumphing over the virus's devastation. However, like other pandemics, COVID-19 significantly impacted the physical and mental health of the afflicted individuals. Our prior study indicated that older age, female gender, unemployment, comorbidities, and hospitalisation were the factors accountable for the deterioration of the individual's QoL [21]. Moreover, our current study pursued these recovered participants to explore the amelioration of their QoL index over time.

In Dhaka city, except for their physical domain, we observed an average rise in psychological, social, and environmental scores among the COVID-19 recovered participants from baseline to the follow-up interview. The individuals' physical domain significantly deteriorated between the two follow-ups, whereas their psychological and environmental overall domain scores witnessed a notable increase. Our findings corroborated the follow-up studies conducted on COVID-19 survivors in Peru and Spain. Even though both groups' QoL drastically declined, the Spanish reported more substantial neurological and psychological damage [28, 29].

However, this exhibited variability across distinct participants' characteristics. Considering the intra-individual disparities between the two interviews, we ascertained that higher age, female gender, history of hospitalisation during COVID-19, and a more significant number of chronic ailments were linked with a lower score in physical, psychological, social, and environmental domains. Conversely, higher education and a higher monthly income were correlated with higher scores, mainly in the psychological domain, followed by other domains. These results were in accordance with our baseline study (6). Another follow-up study of Italian patients with acute respiratory distress syndrome (ARDS) revealed comparable observations [30]. In contrast to a study in Pakistan [31] that reported an improvement in physical QoL over six months after diagnosis, we identified an overall decline in physical QoL over a prolonged period. Despite the reduction in overall QoL score in the physical domain, which was offset by the score increase in other domains, each patient experienced either an increase, decrease or no change in QoL scores between the first and second interviews. Consequently, we selected participants who experienced a decline in their scores and investigated the determinants of the decrease through multivariable regression.

We observed a significant deterioration in the physical and social domain scores among middle-aged to elderly participants during their follow-up interviews. In addition, higher age was associated with a lower physical and psychological domain score. We also discovered that the probability of a reduction in social domain scores increased with a higher age of respondents. This finding is relevant to various studies conducted on different populations during the COVID-19 period [32–38]. This may be because as people age, they experience a decline in their physical health, ultimately constraining their capacity to participate in social activities and maintain relationships. Moreover, older adults may confront social isolation and loneliness, along with low-income family support, which can adversely affect their mental health and overall well-being [39–41]. Another potential factor is that as people age, they may experience a loss of social networks due to retirement, relocation, or the demise of friends and family members

[42, 43]. Finally, age-related changes in sexual function and interest may also contribute to a decline in the social domain of QoL among elderly COVID-19 recovered patients [44, 45].

Our investigation found that employed individuals had a superior QoL with more excellent social stability than their unemployed counterparts 18 months after their recovery from COVID-19. The adverse effects of unemployment on social well-being have been well-established by prior studies [33, 46]. Studies reveal that unemployment is linked to increased social isolation, reduced social support, and decreased access to resources that support social activities [47, 48]. Additionally, the financial strain associated with unemployment may also contribute to reduced QoL as they are responsible for supporting their family during the post COVID-19 period. Unemployed individuals may undergo financial pressure and limited access to resources that can help with social activities, such as leisure activities or transportation. This can exacerbate feelings of social isolation and reduce overall well-being [49]. Furthermore, access to health services was hindered by these cost constraints. A scarcity of affordable health services frequently results from limited resources, exacerbating further restrictions. This intricate interplay between unemployment, financial hardship, social isolation, and inadequate healthcare accessibility underscores the multifaceted challenges arising as detrimental consequences of the COVID-19 pandemic [50].

It was noted that participants with a monthly income greater than 60000 BDT were less likely to have a good physical and psychological QoL than those with a monthly income less than 20000 BDT. This result contradicts surveys conducted on American and Malaysian populations [51, 52]. The plausible reason for this finding is that, following their recovery from COVID-19, individuals with a higher monthly income may have taken on more stressful jobs with longer working hours and less time for self-care, leading to poorer physical health. Additionally, higher-income individuals may have a higher prevalence of sedentary lifestyles, resulting in less physical activity and lower HRQoL [53]. Psychological HRQoL could also be impacted by the stress and pressure associated with individuals' higher monthly family income [54–56]. These individuals may experience higher anxiety, depression, and other mental health issues that could affect their overall QoL. Furthermore, higher-income individuals may have a greater sense of isolation and loneliness, as they may spend more time at work to recover their financial stability and have fewer opportunities for social interaction.

Our longitudinal investigation observed that the QoL of wedded participants was higher than that of unmarried ones. However, more than two-fifths of participants experienced a decline in all four domains of QoL, and married participants were significantly more likely to suffer the decline. One plausible explanation for this discrepancy is that the initial advantage of being married in terms of QoL might have resulted from a spouse's social support and companionship, which might have been particularly vital during the acute phase of convalescence from COVID-19. However, as time elapsed and the acute phase of the illness subsided, the significance of social support and companionship might have declined, resulting in a minor advantage for married participants. Furthermore, the personal relationships, as explored in the social domain, of married participants were affected by other factors such as financial stability, family responsibilities, and caregiving duties, which may have become more noticeable in the long term. Conversely, unmarried respondents may have experienced more support from friends and were satisfied with their personal relationships with their broader social networks during the recovery phase of COVID-19, contributing to a more significant enhancement in their QoL over time. Further inquiry may be necessary to comprehend the reasons behind these findings and identify potential interventions to enhance the QoL of COVID-19-recovered patients, irrespective of their marital status.

In the current study, we found that participants with comorbidities experienced a reduction in physical QoL lower than that of non-comorbid respondents during follow-up interviews.

As comorbid individuals recovered from COVID-19, their average increase in physical QoL might have been higher than that of non-comorbid participants, leading to a lower reduction estimate.

In our investigation, we discovered that COVID-19 immunisation had a profound impact on ameliorating people's psychological QoL after an extended period of 18 months post-recovery from the dangerous disease. At the outset of the pandemic, COVID-19 vaccines played a pivotal role in stemming the fatality rate of this lethal virus. Although social distancing and isolation measures decelerated the transmission of the virus and flattened the epidemic curve in several nations, only vaccination was efficacious in curtailing the spread of the infection, especially in densely populated countries. Individuals who received the COVID-19 vaccine, and those who had already contracted the disease, seem to be safeguarded for a minimum of six months and typically for a more extended duration following their primary series of COVID-19 vaccination resulting in mental relief from the continued stress. Thus, those who refrained from receiving the vaccine encountered negative sentiments about contracting COVID-19 anew, which impinged their self-esteem and lowered their QoL compared to those inoculated. The investigations on COVID-19-afflicted persons and married women residing in rural Bangladesh substantiated that the prospect of reinfection elicited considerable mental stress among the research participants [57, 58].

Finally, it was observed that individuals who suffered from a recurrence of COVID-19 during the interlude between the two follow-up assessments exhibited a higher probability of experiencing a decline in their social QoL scores than their non-reinfected counterparts. This discovery was consistent with research conducted by the Department of Veteran Affairs in the United States, which demonstrated that patients who were infected with COVID-19 multiple times displayed a greater susceptibility to complications in various organ systems, as well as a greater likelihood of being diagnosed with long COVID than those infected only once [59–61]. Remarkably, these outcomes were independent of vaccination status.

This investigation is not without its limitations. Firstly, it is with regret that we could not compare the QoL scores of those who have not been infected with COVID-19 to those who have recovered from the disease. However, prior studies conducted amongst healthy Bangladeshi adolescents and adults between 2005 and 2007 reported an average QoL score of 80–90, in contrast to our study's finding of lower mean QoL scores among participants [62, 63]. While an improvement in QoL would be expected after recovering from COVID-19, our study's divergent results may reflect the adverse socioeconomic impacts of the pandemic on the country's population. Nevertheless, drawing realistic conclusions about this situation is only possible with a control group. Secondly, a considerable number of participants were lost during the follow-up phases. Thirdly, an investigation into the impact of socio-cultural determinants such as healthcare accessibility, financial stability, rehabilitative measures, and health-seeking behaviour on QoL could not be conducted. Fourthly, the repercussions of persistent and debilitating symptoms after COVID-19 were not explored. Nonetheless, our study was one of the few that reported on the QoL of COVID-19 patients over an extended period, highlighting potential implications for policy-level strategies to prevent further deterioration and promote the complete recovery of these individuals.

## Conclusion

In short, the QoL of individuals who have recuperated from COVID-19 has exhibited an amelioration over a span of 18 months, particularly in the psychological, social, and environmental spheres. Nevertheless, several factors, such as the respondent's age, employment status, monthly income, marital status, smoking habits, pre-existing health conditions, vaccination

status against COVID-19, and the possibility of COVID-19 reinfection, were found to have a significant negative association with QoL. To alleviate this situation, policymakers should adopt early diagnosis and prompt management of preventive measures to mitigate the risk of further illness. Psychological and social support should also be advocated for those who experience a reoccurrence of COVID-19. Furthermore, there is a need to undertake action research into QoL in Bangladesh and the long-term impact of COVID-19.

## Supporting information

**S1 Fig. Study participants selection diagram.**
(DOCX)

**S2 Fig. Pattern of change in score in physical, psychological, social, and environmental domains of quality of life.**
(DOCX)

**S3 Fig. Onset of new chronic disease and percentage of re-infection among the recovered COVID-19 participants during second follow-up.**
(DOCX)

**S1 Table. Comparison of quality of life between baseline and follow-up interviews in relation to presence or absence of individual chronic diseases.**
(DOCX)

**S1 Questionnaire.**
(PDF)

**S1 Checklist. PLOS's questionnaire on inclusivity in global research.**
(DOCX)

**S1 Data. Data file.**
(XLSX)

## Author Contributions

**Conceptualization:** Md. Utba Rashid, Koustuv Dalal, Md. Abdullah Saeed Khan, Mohammad Delwer Hossain Hawlader.

**Data curation:** Md. Utba Rashid, Md. Abdullah Saeed Khan, Soumik Kha Sagar, Tahmina Zerin Mishu, Mohammad Hayatun Nabi, Mohammad Delwer Hossain Hawlader.

**Formal analysis:** Md. Utba Rashid, Md. Abdullah Saeed Khan, Umme Kulsum Monisha, Mohammad Hayatun Nabi.

**Methodology:** Md. Utba Rashid, Koustuv Dalal, Md. Abdullah Saeed Khan, Umme Kulsum Monisha, Soumik Kha Sagar, Tahmina Zerin Mishu, Mohammad Hayatun Nabi.

**Supervision:** Mohammad Delwer Hossain Hawlader.

**Validation:** Mohammad Hayatun Nabi.

**Writing – original draft:** Md. Utba Rashid, Koustuv Dalal, Md. Abdullah Saeed Khan, Umme Kulsum Monisha, Soumik Kha Sagar, Tahmina Zerin Mishu, Mohammad Hayatun Nabi, Mohammad Delwer Hossain Hawlader.

**Writing – review & editing:** Koustuv Dalal, Mohammad Delwer Hossain Hawlader.

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
