## [Decision Letter · Decision Letter 0]

18 Jan 2024

PGPH-D-23-02307

Predictors of Health-Related Quality of Life for COVID-19 survivors living in Dhaka, Bangladesh: a repeated Follow-Up after 18 Months of their recovery

Dear Dr. Dalal,

Thank you for submitting your manuscript to PLOS Global Public Health. After careful consideration, we feel that it has merit but does not fully meet PLOS Global Public Health’s publication criteria as it currently stands. Therefore, we invite you to submit a revised version of the manuscript that addresses the points raised during the review process.

We look forward to receiving your revised manuscript.

Kind regards,

Jianhong Zhou

Staff Editor

Journal Requirements:

1. Please include a complete copy of PLOS’ questionnaire on inclusivity in global research in your revised manuscript. Our policy for research in this area aims to improve transparency in the reporting of research performed outside of researchers’ own country or community. The policy applies to researchers who have travelled to a different country to conduct research, research with Indigenous populations or their lands, and research on cultural artefacts. The questionnaire can also be requested at the journal’s discretion for any other submissions, even if these conditions are not met.  Please find more information on the policy and a link to download a blank copy of the questionnaire here: https://journals.plos.org/globalpublichealth/s/best-practices-in-research-reporting. Please upload a completed version of your questionnaire as Supporting Information when you resubmit your manuscript.”

2. In the ethics statement in the Methods, you have specified that verbal consent was obtained. Please provide additional details regarding how this consent was documented and witnessed, and state whether this was approved by the IRB.

3. Please provide separate figure files in .tif or .eps format only and remove any figures embedded in your manuscript file. Please also ensure all files are under our size limit of 10MB.

Additional Editor Comments (if provided): We note that one or more reviewers has recommended that you cite specific previously published works. As always, we recommend that you please review and evaluate the requested works to determine whether they are relevant and should be cited. It is not a requirement to cite these works. We appreciate your attention to this request.

Reviewers' comments:

Reviewer's Responses to Questions

**Comments to the Author**

1. Does this manuscript meet PLOS Global Public Health’s publication criteria? Is the manuscript technically sound, and do the data support the conclusions? The manuscript must describe methodologically and ethically rigorous research with conclusions that are appropriately drawn based on the data presented.

Reviewer #1: Yes

Reviewer #2: Yes

2. Has the statistical analysis been performed appropriately and rigorously?

Reviewer #1: Yes

Reviewer #2: Yes

3. Have the authors made all data underlying the findings in their manuscript fully available (please refer to the Data Availability Statement at the start of the manuscript PDF file)?

Reviewer #1: Yes

Reviewer #2: Yes

4. Is the manuscript presented in an intelligible fashion and written in standard English?

Reviewer #1: Yes

Reviewer #2: No

5. Review Comments to the Author

Reviewer #1: First of all Thank you for giving me this opportunity.

the item is acceptable. However, I leave it up to you to evaluate the conformity of the work with the magazine's guidelines

The article is acceptable given the great interest of the topic for the scientific community.

the article touches on a very delicate and complex topic. However, the authors clearly described the results and discussion.

Reviewer #2: Dear Authors, thank you for giving me the opportunity to read your study. In a large cohort of COVID-19 patients, you found some risk factors for worse QOL or some protective factors for better QOL:

The study in interesting, and Authors should be commended for the effort.

I have some suggestions that I hope they will improve the quality of your paper.

First, I recommend English proofreading because some parts are difficult to read and understand.

Second, introduction is very/too long. Please, after a general overiview of the problem, focus on what you will investigate in your study.

Third, one factor that could highly impair QOL after COVID infection is the admission to the ICU. In this regard, if you feel appropriate, consider this article: J Clin Med. 2023 Jan 29;12(3):1058. doi: 10.3390/jcm12031058. PMID: 36769705; PMCID: PMC9918008.

As you correctly pointed out, one of the main problems during pandemic was the limited resources especiallt in low income countries.

You find interesting insights in this nice and appropriate in my opinion article: Acta Biomed. 2021 May 12;92(2):e2021097. doi: 10.23750/abm.v92i2.11159. PMID: 33988143; PMCID: PMC8182622.

Discussion: please, try to reformulate this section giving more emphasis to your results. In other words, it is right to sum up literature evidence, but is also important to highlight your results.

Hope my comments will be useful.

Best regards

6. PLOS authors have the option to publish the peer review history of their article (what does this mean?). If published, this will include your full peer review and any attached files.

**Do you want your identity to be public for this peer review?** For information about this choice, including consent withdrawal, please see our Privacy Policy.

Reviewer #1: No

Reviewer #2: No

---

## [Decision Letter · Decision Letter 1]

21 Jun 2024

Predictors of Health-Related Quality of Life for COVID-19 survivors living in Dhaka, Bangladesh: a repeated Follow-Up after 18 Months of their recovery

PGPH-D-23-02307R1

Dear Professor Dalal,

We are pleased to inform you that your manuscript 'Predictors of Health-Related Quality of Life for COVID-19 survivors living in Dhaka, Bangladesh: a repeated Follow-Up after 18 Months of their recovery' has been provisionally accepted for publication in PLOS Global Public Health.

Best regards,

Gautam I Menon, PhD

Academic Editor

Reviewer Comments (if any, and for reference):

Reviewer's Responses to Questions

**Comments to the Author**

1. If the authors have adequately addressed your comments raised in a previous round of review and you feel that this manuscript is now acceptable for publication, you may indicate that here to bypass the “Comments to the Author” section, enter your conflict of interest statement in the “Confidential to Editor” section, and submit your "Accept" recommendation.

Reviewer #2: All comments have been addressed

Reviewer #3: All comments have been addressed

2. Does this manuscript meet PLOS Global Public Health’s publication criteria? Is the manuscript technically sound, and do the data support the conclusions? The manuscript must describe methodologically and ethically rigorous research with conclusions that are appropriately drawn based on the data presented.

Reviewer #2: Yes

Reviewer #3: Yes

3. Has the statistical analysis been performed appropriately and rigorously?

Reviewer #2: Yes

Reviewer #3: Yes

4. Have the authors made all data underlying the findings in their manuscript fully available (please refer to the Data Availability Statement at the start of the manuscript PDF file)?

Reviewer #2: Yes

Reviewer #3: (No Response)

5. Is the manuscript presented in an intelligible fashion and written in standard English?

Reviewer #2: Yes

Reviewer #3: Yes

6. Review Comments to the Author

Reviewer #2: Paper has improved

Reviewer #3: This is an important study, highlighting the interlink between quality of life and COVID-19 and other factors. The findings and recommendations could be used to inform appropriate response strategies.

I would like to congratulate authors for a well-written paper.

7. PLOS authors have the option to publish the peer review history of their article (what does this mean?). If published, this will include your full peer review and any attached files.

**Do you want your identity to be public for this peer review?** For information about this choice, including consent withdrawal, please see our Privacy Policy.

Reviewer #2: No

Reviewer #3: No
